# Stochastically perturbed physics-tendencies based ensemble mean approach in the WRF model: a study for the North Indian Ocean tropical cyclones

Gaurav Tiwari, Vishal Bobde, Pankaj Kumar, and Alok Kumar Mishra

Department of Earth and Environmental Sciences,
Indian Institute of Science Education and Research Bhopal, Bhauri-462066, India

*Correspondence to*: Pankaj Kumar (kumarp@iiserb.ac.in)

## Abstract

Tropical cyclones (TCs) are among the catastrophic natural hazards over the North Indian Ocean (NIO), and they are expected to become more frequent in the upcoming years. TCs occur primarily in the pre-monsoon (April-June) and post-monsoon (October-December) seasons, wreaking havoc on South Asian regions. For reliable alerts and disaster warnings ahead of time, better forecasting of TC features such as track, landfall, intensity, rainfall, and so on is crucial. The present study uses the stochastically perturbed physics-tendencies

(SPPT) ensemble-mean approach along with digital filter initialization (DFI) to the initial and boundary conditions for the high-resolution Weather Research and Forecasting (WRF) model. The model's sensitivity has been investigated for the two NIO TCs, Tauktae (in May 2021) and Nivar (in November 2020), by performing a large number of experiments. Compared with control runs, the track simulations in terms of the reduction in along-track (cross-track) errors for Tauktae and Nivar were improved by 68.8% (23.4%) and 28.2% (40.7%),

respectively, in the DFI experiment. Further improvements were found in the SPPT-based ensemble mean experiments (DFI+SPPT) as the along-track (cross-track) errors, compared to control simulations, were reduced by 65.3% (27.7%) and 37% (54.1%), for Tauktae and Nivar, respectively. However, the DFI simulations showed a potential to improve the TCs' track simulation but failed to reduce the error in intensity simulation. On the other hand, DFI+SPPT experiments improved the model's reliability in simulating TCs' intensity (maximum

sustained wind speed and minimum sea-level pressure) considerably. Thus, the DFI+SPPT experiments showed higher skills in simulating the TCs' characteristics.

Keywords: North Indian Ocean; Tropical Cyclone; WRF; SPPT; DFI

## 1    Introduction

Tropical cyclones (TCs) are the most catastrophic natural disasters, posing a severe threat to many aspects of

life and property, as well as agricultural and industrial productivity, regardless of whether or not they make landfall, resulting in substantial socio-economic consequences. According to WMO technical report 2008, the north Indian Ocean (NIO), which includes the Bay of Bengal (BoB) and the Arabian Sea (ARB), generates about 7% of worldwide TCs (Osuri et al., 2012). The NIO, particularly the ARB, is projected to encounter frequent and intense TCs (Deshpande et al., 2021) as the earth continues to warm, increasing the consensus.

Every year, in the BoB and ARB basins, intense TCs strike, mainly in the months of April-June (pre-monsoon)





and October-December (post-monsoon), with the latter being more dominant (Tiwari et al., 2021). A better prediction of TC attributes (such as track, landfall, severity, rainfall, and so on) is crucial to provide precise alerts, disaster warnings, and mitigation efforts 48-72 hours ahead of time (Dodla et al., 2016). Coastal urbanization and industrial activity have significantly increased TC economic losses during the last decade or

two. However, developments in numerical weather prediction (NWP) systems have resulted in a significant drop in the number of people who died recently (Dodla et al., 2016; Kanase and Salvekar, 2014).

For the past several decades, high-resolution NWP models have played a vital role in weather forecasting. With the effective implementation of some advanced techniques and the application of various limitations, such as initial conditions, domain, parameterization, and so on, the reliability of NWP prediction skill has been

qualitatively upgraded (Chandrasekar et al., 2012). Models can produce different results with different combinations of physics and parameterization schemes (Mishra and Dubey, 2021). Additionally, model resolutions also play a crucial role in prediction skills. The high-resolution helps resolve the complex topography and sub-grid processes (Mishra et al., 2021). The NWP model performances are affected by grid sizes and the geographical region of interest; hence, the best physical parameters cannot be identified

universally for every region of the globe. The best setup for a region might be the worst for others. Therefore, careful selection and tuning of the parameters are needed before finalizing a model for a respective region that strongly influence the prediction of tropical cyclones.

Efforts are being continued to improve the NIO tropical cyclone prediction by NWP models (Panda and Giri, 2012). Mohanty et al. (2010) compared the relative performances of the three input forcings, namely Global

Forecast System (GFS), Final Analysis (FNL), and National Centre for Medium-Range Weather Forecasting (NCMRWF) in the weather forecasting and Research (WRF) for NIO TCs simulation. They reported that the FNL reanalysis dataset produced the best rainfall and track prediction results. Singh and Mandal (2014) did a case study on cyclone Viyaru over the BoB basin with GFS and FNL dataset, and the result showed that FNL is consistently better with track and intensity prediction. Srinivas et al. (2013) found the WRF model to have the

best predictions for north-westward moving cyclones. However, the WRF model with high-resolution can capture the recurving cyclone tracks and intensity over NIO (Osuri et al., 2013). Kumar et al. (2020) simulated the two ARB TCs, Chapala and Megh, using the WRF model with different initializations and found reasonable results in TC's landfall location.

Considering TCs an initial value problem, the initialization process highly influences the prediction skill. A

small error in the initial condition deteriorates the NWP model's prediction skill with time. Furthermore, the prediction is affected by imbalances among initial mass, wind fields and noises, such as inertial gravity waves (Peckham et al., 2016). Thus, their minimization is a very crucial step in the NWP simulations. These inertial gravity waves contaminate the short-range forecast and TC track and intensity in NWP model simulations. However, by incorporating digital filter initialization (DFI; Lynch and Huang 1992) into the NWP system, these

noises can be suppressed to some extent, potentially resulting in a better model forecast. In addition, ensemble approaches have been applied in NWP modelling systems (Zhu, 2005). As a result, the forecast ensemble's mean can predict better than the control forecast.

A very few studies (e.g., Leutbecher et al., 2017; Puri et al., 2001) have investigated the effectiveness of stochastic physics to generate the ensemble spread and eventually to improve the NWP model's skill. Puri et al.





(2001) found that the stochastic physics approach improved the intensity simulation of Western Pacific and Atlantic regions' typhoons but showed less impact on track simulation. Therefore, in this study, the sensitivity of the initialization is investigated using DFI and stochastically perturbed physics-tendency (SPPT) based ensemble-mean techniques to simulate the characteristics of two NIO TCs, Tauktae (in May 2021) and Nivar (in November 2020). Synoptic conditions of the selected cases are depicted in section 2, model description, data

and methodology are described in section 3, results are provided and discussed in sections 4 and 5, followed by conclusions in section 6.

## 2 Description of the selected cyclones

### 2.1 ESCS Tauktae (2021) over the ARB in the pre-monsoon season

On May 13, 2021, an extremely severe cyclonic storm Tauktae developed from a low-pressure area in the south-

eastern region of the ARB (Fig. 1) (ESCS Tauktae Report IMD, 2021; RSMC New Delhi Report, 2021). On May 14, this system concentrated into a depression due to the favourable conditions. At midnight on May 14, it morphed into a cyclonic storm (CS) due to its rapid evolution. It travelled parallel to India's west coast for a few days, deepening into a severe cyclonic storm (SCS) and a very severe cyclonic storm (VSCS). It became an ESCS on May 16 at noon, with rapid intensification. Tauktae made landfall over the Saurashtra coast in Gujarat

state on the same evening, changing its path to the northeast.

The Effect of Tauktae can be seen along the entire west coast of India as it travelled parallel to the coast. It rapidly intensified from 16th to 17th May, gaining about 18 m/s sustained wind speed over 24 hours. Tauktae had a 129 hours lifespan with 12 hourly mean translational speed of 14.4 km/h, which was more than the average value (11.8 km/hour) of very to extremely severe cyclonic storms' translational speed over time the

ARB during pre-monsoon season. A massive storm surge was observed in the coastal districts of Saurashtra around the landfall time, and more than 120 people lost their lives with the highest fatalities of 67 in the state of Gujarat. Approximately 1.1 million people were affected in 421 villages, and over 19,500 houses were damaged (ESCS Tauktae Report IMD, 2021).

### 2.2 VSCS Nivar (2020) over the BoB in the post-monsoon season

On November 22, 2020, a very severe cyclonic storm Nivar emerged from a low-pressure area in the BoB's south-central region (Fig. 1) (VSCS Nivar Report IMD, 2020; RSMC New Delhi Report, 2020). Later that day, it concentrated into depression. Cyclone Nivar moved north-westward in its overall path. In the evening of November 24 and the morning of November 25, the system strengthened into a SCS and a VSCS, respectively. On the same day, it made landfall near Puducherry. As the storm weakened after landfall, it shifted northward,

eventually curving slightly eastward towards the end. On the evening of the 26th, Nivar dissipated into depression in Andhra Pradesh, completing its life cycle.

VSCS Nivar had a peak intensity of 33.4 m/s around 12 UTC of 25[th] November, and it didn't show rapid intensification. Nivar has a life period of about 93 hours and relatively less 12 hourly mean translational speed of 8.9 km/h compared to the average 12 hourly translational speed (15 km/hour) of VSCS over the BoB during

the post-monsoon season. Heavy rainfall of 7-12 cm/day associated with Nivar was observed along the coast of

Tamil Nadu and southern Andhra Pradesh on 26th and 27th November. No significant storm surge was observed, and eight causalities were reported in Andhra Pradesh. Over 112,000 people were affected by cyclone; about 2,618 small animals, 88 large animals and 8,130 poultry birds were reported dead. Thirty-four thousand hectares of standing crops were damaged in the Prakasam district and 2500 hectares of paddy

seedlings drowned due to this cyclone (VSCS Nivar Report IMD, 2020).

### 3    Model description and data

This study used a fully compressible non-hydrostatic Advanced Research WRF model developed by the National Center for Atmospheric Research. For TCs Tauktae and Nivar, the WRF model (version 4.2) was configured with 9 km horizontal resolution and 42 vertical levels over the ARB and BoB, respectively. The

model's initial and boundary conditions were generated using $0.25° \times 0.25°$ reanalysis data from the FNL and GFS (Table 1). The land surface boundary conditions were obtained from the United States Geological Survey with a horizontal grid spacing of 5 minutes. The model's predictive variables include potential temperature, geopotential, surface pressure, perturbation quantities, three-dimensional wind, turbulent kinetic energy, and scalars like cloud water and water vapour mixing ratio. The planetary boundary layer physics used in the present study

is Yonsei University scheme with WRF single-moment 3-class microphysics scheme, Grell 3D Ensemble (Kain-Fritsch) cumulus convection scheme for Nivar (Tauktae), Rapid Radiation Transfer Model (RRTM) scheme for long-wave radiation, and Dudhia scheme for shortwave radiation. Model outputs for track and intensity prediction were compared using IMD best-track, Advanced Scatterometer (ASCAT), the European Centre for Medium-Range Weather Forecasts Reanalysis 5th Generation, and the Modern-Era Retrospective analysis for

Research and Applications Version 2 (MERRA2) datasets, while simulated rain's validation was conducted using Multi-Source Weighted Ensemble Precipitation (MSWEP) data.

### 3.1 Experimental setup and methodology

In this study, a set of experiments have been performed (Fig. 2).

 i) *Control simulations (CNTL):* In which real forcings (initial and boundary) were used without any filtering

and perturbation

ii) *DFI simulations:* The simulation in which a twice DFI low-pass filter scheme (Lynch and Huang, 1994) available in the WRF model was used, and

iii) *DFI+SPPT simulations:* A random perturbation was applied to the tendencies from the convection and planetary boundary layer schemes (Berner et al., 2015) allied with DFI.

SPPT perturbs the physics tendencies with a random multiplicative coefficient depending on the uncertainty at each grid point and timestep. The temperature, humidity, and wind components are perturbed using SPPT (Skamarock et al., 2021). The setting for SPPT namelist parameters were adopted from Li et al. (2020) given in Table 1:

Since the driving forcings significantly control the model's performance; thus, it is crucial to obtain the best

performing setup in the combination of model parameters. Sensitivity experiments were performed to tune the



model, and after getting the reliable set, we demonstrated the model's sensitivity to two different driving data, namely, FNL and GFS. The model simulations with FNL forcing showed better results than GFS. Therefore, this study utilized the initializations from the FNL data to examine the impact of DFI+SPPT. A detailed description is provided in the supplementary file. Forcings with initialization time 14_06 and 22_18 was used

for ESCS Tauktae and VSCS Nivar, respectively. The details of CNTL, DFI, and DFI+SPPT are described above. A fixed "Best Member" among all ten ensemble members has also been examined throughout the analysis, and its selection was based on the TCs' trajectory.

We have compared the model results for the TC's track simulation from the IMD best-track data in terms of the cross-track (CT) and along-track (AT) errors. The calculation method of CT and AT error is provided in Fig. 3.

Curves OA and O'B represent the TC track from the observation and model output. At some time "t", the cyclone position from observation (model) was at point A (B), whereas points O (O') denoted the TC's location 6-hour before at time "t-1". A perpendicular from B to the straight-line OA meets at point B. Thus, the distance BP denotes a CT error in the model output. It is also noted that the AT and CT error cannot be calculated for the very first integration time step from the model simulation. A positive and negative CT error shows the right and

left forecast positions, respectively. Similarly, the distance AP denotes an AT error where positive (negative) signs represent the cyclone's faster (slower) movement.

## 4 Comparative assessment of model simulations with observation

### 4.1 Trajectories of the simulated cyclones

The simulated tracks from all the experiments along with observation for Tauktae and Nivar are shown in Figs.

4(a-b). The best ensemble members (here after "Best Member") from DFI+SPPT simulations are also taken into account based on their trajectories, and the same Best Member were used throughout the analysis.

From all the experiments (i.e., control simulation, DFI, DFI+SPPT, and Best Member) of Tauktae, the initial vortices of the storm were found reasonably close to the observation. Simulated tracks showed a drift to the left of the observation after first few integration time-steps (6-hourly) (Fig. 4a). The control experiment had the

farthest track among all. The discrepancy was minimised in both tuned experiments, DFI and DFI+SPPT, and approaches closer to observation (Fig. 4a). The Best Member track was closest to observation till landfall compared to others; whereas, DFI+SPPT outperformed DFI. On the other hand, the TC's genesis area and initial movement from model experiments were predominantly right to the observation for Nivar (Fig. 4b). The trajectories from the DFI, Best Member, and DFI+SPPT experiments crossed the observed path a few times and

made landfall in good agreement with it compared to control run. As expected, the ensemble members produced a wide range of landfall locations. The DFI and DFI+SPPT experiments improved the trajectories of selected TCs in visualization. Further we examined the reduction in along-track and cross-track errors to get more robust results.

### 4.2 Along-track and cross-track errors in simulated tropical cyclones track detection

For Tauktae (Fig. 5a), the model exhibited substantial AT errors (approximately 100 km) from CNTL and DFI+SPPT in the early hours (at 6-hour mark), whilst DFI and Best Member were attributed with lesser errors





(nearly 70 km). Except for a few instances, the cyclone's propagation from model simulations was generally behind the observation (24- and 30-hour marks). Furthermore, with the passage of time, more improvements were found. The DFI was detected with the least error near the landfall (at the 78-hour mark), followed by relatively some more error from the DFI+SPPT and Best Member. After the landfall, control run performed poorly and revealed significant errors. The DFI (24.98 km) had the minimum mean absolute error (MAE) over the whole life of TC, closely followed by DFI+SPPT (27.83 km) and Best Member (31.41 km), and the CNTL had the maximum MAE (80.18 km). As a result, there was an improvement of 68.8 % (65.3 %) in DFI (DFI+SPPT) concerning CNTL.

In contrast to Tauktae, Nivar's AT error size was smaller initially and increased at 24-30 hours in all experiments. The error was significantly reduced from 36 hours to landfall, with DFI producing the lowest error (30.04 km), closely followed by DFI+SPPT (34.2 km) (between 72-78 hours). Overall, the DFI+SPPT (40.05 km), Best Member (41.28 km), and DFI (45.67 km) had less MAE, while the CNTL had the maximum MAE (63.6 km). As a result, when compared to CNTL, the AT error in the DFI (DFI+SPPT) improved by 28.2 % (37.03 %).

In comparison to along-track, the cross-track error in Tauktae showed a different distribution with time for all the experiments (Fig. 6a). It was smaller in the initial hours and gradually increased even after the landfall. Before the TC's landfall (78 hours), the CT error was found to be minimum (maximum) from the Best Member (CNTL), whereas DFI and DFI+SPPT closely matched with Best Member. The overall MAE from CNTL (126.98 km), DFI (97.27 km), DFI+SPPT (91.81 km), and Best Member (75.85 km), indicate an improvement of 23.4%, 27.7%, and 40.3% in the DFI, DFI+SPPT, and Best Member simulations experiments compared to CNTL, respectively.

In the case of Nivar, simulated tracks crossed the observed track two-three times in the first 48 hours, and no monotonic pattern in the CT error was seen from any experiment. However, from 54 hours onward, all the experiments exhibited the tendency of a gradual decrease in the error. During 54 hours till the landfall (between 72 and 78 hours), the error size was approximately two to three-fold in the CNTL than tuned experiments (DFI, DFI+SPPT, and Best Member). Similar to Tauktae, the MAE from CNTL (23.29 km), DFI (13.82 km), DFI+SPPT (10.7 km), and Best Member (10.3 km) indicate an improvement of 40.7%, 54.1%, and 55.8% in the DFI, DFI+SPPT, and Best Member experiments compared to CNTL, respectively.

The above results indicate that implementing the digital low-pass filter had significantly reduced the AT and CT error in the selected TCs' prediction, which was further reduced in the DFI+SPPT led to the most robust experiment amongst all.

### 4.3 Intensity of the simulated cyclones

Forecasting TC's intensity is crucial for minimising TC-related casualties and damages. The minimum sea level pressure (MSLP) and maximum sustained wind (MSW) speed are the two primary and widely accepted parameters that are used to address the TC's intensity (Tiwari et al., 2021). Hence, the analysis has been conducted to examine the prediction of selected TCs' intensity signified by 6-hourly MSLP (MSLP error) and





MSW (MSW error) speed in Figs. 7 (8) and 9 (10), respectively. The following sub-sections go over detailed evaluation.

### 4.3.1 Improvement in MSLP simulation

The time evolution of MSLP is depicted in Fig. 7. It can be seen from Fig. 7a that all the experiments showed overestimation with varying magnitude for Tauktae. The DFI, DFI+SPPT, and Best Member experiments captured the intensification and weakening stage much better than CNTL, which kept on intensifying even after the landfall (between 78 and 84 hours) also. The lowest MSLP (950 hPa) in the IMD was observed at 66-72 hours, followed by a 6–12-hour delay in the DFI (943 hPa), DFI+SPPT (948 hPa), and Best Member (948 hPa), and an 18–24-hour lag in the CNTL (941 hPa).

All experiments for Nivar (Fig. 7b) showed an overestimation tendency with heterogeneity in their intensification and weakening phases. Their evolution was closely propagated with observation during the first 24 hours, and then the overestimation occurred drastically. The control and DFI showed a similar pattern of intensification and decay phases concerning observation, while the Best Member attained its peak intensity (lowest MSLP) 6-hour earlier. Unlike others, the DFI+SPPT reached its peak intensity 12-18 hours earlier to observation; consequently, the intensity was relatively well captured by this experiment compared to others near the landfall.

To make the result more perspective, we demonstrated the comparative performance of all experiments by computing the error/bias (observation minus model) in MSLP (Fig. 8). For ESCS Tauktae, all experiments showed the tendency of increasing error gradually for the first 30 hours and then decreased till 60 hours (Fig. 8a). The error reduced drastically for the next 12 hours, showing overestimation in the DFI and underestimation in others. Around the landfall, the CNTL and DFI produced larger errors, whereas DFI+SPPT and Best Member have shown the lesser errors. The absolute mean MSLP error from all four experiments, CNTL, DFI, DFI+SPPT, and Best Member, were 15.95, 16.64, 11.29, and 13.19 hPa, respectively. Thus, MSLP simulation improved by 29.2% (17.3%) in DFI+SPPT (DFI), with a minimal retrogression of 4.3% in DFI compared to CNTL.

For the case of Nivar, the MSLP error from all the experiments had shown a gradual increase from initial hours till the landfall (between 72 and 78 hours) except DFI+SPPT, which showed a decreasing pattern after 54-hour forecast time (Fig. 8b). Like Tauktae, the MSLP error near the landfall was minimum (maximum) in DFI+SPPT (DFI). The absolute mean MSLP error from all four experiments, CNTL, DFI, DFI+SPPT, and Best Member, were 8.67, 10.4, 7.57, and 9.07 hPa, respectively. Therefore, MSLP simulation improved by 12.7% in DFI+SPPT and diminished by 4.6% (19.9%) in Best Member (DFI) compared to CNTL. The above discussion indicates that the DFI+SPPT have produced the best results to simulate cyclone related MSLP relative to others.

### 4.3.2 Improvement in MSW simulation

In addition to MSLP, MSW is the key parameter that determines the cyclones's intensity, and Fig. 9 depicts the time evolution of MSW (m/s). The MSW for Tauktae reacted differently than MSLP. Model simulations of Tauktae showed overestimation early in TC's life, underestimation during the landfall phase, and then overestimation after the landfall, with variability in intensity and weakening stages (Fig. 9a). The DFI+SPPT



(40.43 m/s) and Best Member (50.13 m/s) reached their peak at 48 hours mark by nearly 6 and 18 hours before DFI (51.4 m/s) and observation (51.44 m/s), respectively, while the CNTL (52.42 m/s) did not reach its peak before the landfall (between 78 and 84 hours).

All of the experiments in the case of Nivar (Fig. 9b) indicated an overestimation; however, DFI+SPPT exhibited a modest underestimation near the landfall phase. The CNTL, DFI, and Best Member simulations followed a very similar pattern throughout the TC's life, while DFI+SPPT followed the same pattern but only for 42 hours before drifting towards the observation. By the way, the DFI+SPPT shifted back to the other experiments after landfall.

The vertical distribution of azimuthally averaged wind speed from the model simulations (Fig. S10 provided in the supplementary file) also agree in the same manner with Fig. 9a. The maxima of wind contours were found between the 50 and 100 km extending from lower to mid and upper troposphere; whereas, in the case of Nivar, the maxima of wind contours were centred around 50-60 km (Fig. S11). Stronger wind contour eventually resulted into a peak of MSW from the particular experiment.

The MSW error (m/s) for both cases in terms of the difference from observation is presented in Fig. 10. For Tauktae, all the experiments showed a positive MSW error for the initial 42 hours (Fig. 10a). From 48 hours till the landfall, some experiments showed negative and positive errors both. Interestingly, the DFI+SPPT experiment produced maximum error near the landfall; however, its overall performance was found to be best. The mean absolute error in control, DFI, DFI+SPPT, and Best Member experiments were 11.64, 10.66, 7.18, and 9.52 m/s showing an improvement of 8.42%, 38.32%, and 18.2% in the DFI, DFI+SPPT, and Best Member experiments, respectively, relative to control run.

For Nivar, all the experiments showed an overestimation except for DFI+SPPT at 66 and 72 hours (Fig. 10b). DFI (DFI+SPPT) experiments produced the worst (best) results for MSW error during the landfall hours. The mean absolute error in CNTL, DFI, DFI+SPPT, and Best Member was 8.54, 9.61, 5.45, and 9.02 m/s showing a deterioration of 12.5% (5.6%) and an improvement of 36.2% in DFI (Best Member) and DFI+SPPT, respectively, compared to CNTL.

Hence, the DFI+SPPT experiment has better simulated the TC's intensity in terms of both MSLP and MSW. The results reveal that, unlike track prediction, the DFI experiment did not significantly improve the intensity prediction of the selected TCs. The DFI+SPPT had a substantial impact on intensity simulation as well.

### 4.4 Cyclone-induced rainfall simulation

Flooding, storm surge, and other risks are inherent with cyclones due to excessive rain. It highlights the significance of precise prediction of TC-induced rain, particularly over low-lying Indian coastal regions. The spatial distribution of TC-induced 24-hour accumulated rain (mm/day) during the maximum intensity of selected TCs (00 UTC 17 May to 00 UTC 18 May 2021; for Tauktae and 00 UTC 25 November to 00 UTC 26 November 2020; for Nivar) is provided in Figs. 11-12 from all the experiments and observation (MSWEP). The IMD best-track for the same period is also overlayed. The observation showed a rain cluster towards the left of the TC's track over the ocean and a significant rainfall over the southern (western) part of states Gujarat (Maharashtra) for Tauktae (Fig. 11e). The CNTL showed that the majority of the rain occurred over the ocean,



leading to two distinct rainfall clusters, inconsistent with the observation (Fig. 11a). DFI showed better precipitation than CNTL; however, the magnitude was slightly overestimated and northward shifted in two rain clusters (Fig. 11b). DFI+SPPT showed substantial improvement by reducing overestimation and producing a single cluster of rain; however, the location of rainfall maxima was slightly southward shifted (Fig. 11c). The amount of rainfall over the southern Gujarat region was also closely aligned with observation. The Best Member also had a single rain cluster that was located near the landfall area (Fig. 11d).

Compared to Tauktae, the model experiments for Nivar produced a higher proportion of overestimation in rainfall prediction (Fig. 12). This overestimation could be due to unresolved updrafts and downdrafts of shallow and convective clouds within selected cumulus scheme sub-grid-scale processes. In the observation, the cluster of maximum rain was along the track slightly shifted towards the left of the TC's propagation (Fig. 12e). More or less similar patterns with excess rain were found in all the experiments considering relatively lesser overestimation in the DFI+SPPT (Fig. 12c). CNTL (Fig.12a) had very high overestimation, which was unreasonable even over the land, and similar patterns were seen in DFI (Fig.12b) and Best Member (Fig.12d).

To find robust conclusions, the comparison of model results with observation for rainfall simulation was further analyzed by using quantitative skill scores, such as bias score and equitable threat score (ETS), for the selected TCs at various threshold ranges (Fig. 13(a-d)). The ratio of locations where the model and observation were both over the threshold is represented by a bias score. ETS is a well-known metric for determining how accurate a forecast is compared to random chance.

Although the bias score for Tauktae (Fig. 13a) was lower in CNTL than DFI, DFI+SPPT, and Best Member in producing lower intensity precipitation (50-200 mm/day), the patterns were more or less opposite at higher threshold values (200-300 mm/day). Furthermore, there was a significant difference in the ETS between CNTL and tuned simulations (Fig. 13b). The ETS for CNTL is lower than other experiments, and it quickly goes into negative territory at about 170 mm threshold. So, the CNTL forecast was found statistically unreliable for predicting any rainfall above that threshold. DFI, DFI+SPPT, and Best Member performed much better with a positive ETS up to 250 mm/day threshold. The DFI+SPPT and Best Member outperformed DFI for all threshold values while DFI+SPPT further performed Best Member also for higher threshold. On the other hand, Nivar had a high bias score in all experiments (Fig. 13c); this can be attributed to overestimation by the model, as seen in Fig. 12. The pattern of bias score and ETS was not much different for all the experiments. The Best Member performed better for low threshold values, whereas DFI+SPPT performed better at higher thresholds. If comparing DFI and DFI+SPPT, DFI+SPPT performed consistently better above the 100 mm/day threshold.

In both cases, the spatial distribution and ETS of model-generated rainfall revealed that the CNTL experiments had high location, spread, and size uncertainty. Despite some slight improvements in DFI, the DFI+SPPT tests yielded the best results, closely matched with the Best Member ensemble component.

## 5 Representation of the selected cyclones from different experiments

### 5.1 Structure of the cyclones

The mature cyclones are comprised of a horizontal quasi-symmetric circulation classified as the primary circulation. The horizontal wind distribution (Figs. 14 and 15) and maximum reflectivity of the cloud indicate





the structure of selected cyclones in different experiments (Figs. 16 and 17). The vectors represent wind flow,
whereas the contours illustrate the magnitude of wind patterns and compared to the ASCAT satellite winds
product. The yellow and green symbols indicate the cyclone's location from the observation (IMD) and model
experiments. The observation (ASCAT) demonstrated north-westerlies over the ARB in the case of Tauktae
(Fig. 14). Although the same north-westerlies were presented in the model simulations, they were determined to
be stronger than observed. However, simulated winds revealed more prominent cyclonic circulation over a
larger region in the ARB, resulting in an overestimation of the cyclone's intensity. In the control experiment, the
cyclone's centre location was southwest of the actual event, and south-easterlies were driving it away from the
observed position. This circulation accelerated the storm's northwest movement and showed the larger
inaccuracy in track detection. Other experiments captured the storm markedly better than the control run. The
Best Member simulation was found to be the best of all, closely followed by the DFI+SPPT and DFI
simulations. In the case of cyclone Nivar (Fig. 15), whilst the ASCAT satellite product identified the storm's
location, the wind magnitude around the eyewall was inadequately demonstrated. The observed scale of
maximum wind (10-18 m/s) was significantly lower than that of model simulations (20-40 m/s) and IMD best-
track (Fig. 9b). However, the cyclone's eye in the different experiments were relatively close to the observation
(yellow dot) than Tauktae. As a result, the development of the stated wind distribution may have shifted the
cyclones in the control run away (close) from the observed path, deteriorating (improving) the model
performance.

The spatial distribution of simulated maximum reflectivity of cloud (in dBZ) from different experiments valid at
peak intensity, 2021-05-17_00 UTC and 2020-11-25_18 UTC for Tauktae and Nivar, respectively, are shown in
Figs. 16 and 17. The black and purple symbols indicate the cyclone's centre location in the observation and
model experiments. The maximum reflectivity signatures were fully resolved for both cyclones as the model
produced a well-defined comma cloud band structure. These bands proxy the deep convection, cloud activity,
and precipitation events. In the case of Tauktae, the control and DFI experiments showed the larger magnitude
(ranges from 40-60 dBZ) in the inner rainband regions (extends from eyewall to 100 km) as well as in the outer
rainband regions (starts from 150-200 km). A relatively weaker magnitude was found in the DFI+SPPT and
Best Member experiments. These distributions were close to the spatial patterns of the rainfall (Fig. 11). On the
other hand, the magnitude of maximum reflectivity was slightly less for the cyclone Nivar (Fig. 17). The model
showed extended comma cloud bands, and maximal (40-55 dBZ) was found to the southwest of the storm's
centre. These patterns were also matched with the rainfall distribution (Fig. 12).

### 5.2 Cyclones' movement

The propagation of selected TCs is explained by accumulated upward latent heat flux at the surface level
(ALHF; $J/m^2$) from all the experiments (Fig. 18a-h). In the case of Tauktae, the initial vortex from CNTL (Fig.
18a) was formed southwest to the observed location, whereas from the tuned experiments, the start point of the
TCs was relatively close to the observation (Fig. 18b-d). Again, from Fig. 18a, a hot spot in the central-east
ARB pulled the cyclonic circulation towards it, and TC followed the westward movement for a few hours and
again travelled north-eastward and then slightly north-westward. This westward movement eventually caused a
significant error in the TC Tauktae's track prediction from the CNTL experiment. In the DFI experiment (Fig.



18b), the magnitude of ALHF was stronger than CNTL, and we can see two hotspots, one in the southwest of the track and another on the east side. This dipole type distribution ultimately restricted the westward movement of Tauktae; consequently, it did not go far away from the actual trajectory resulting in less error in the track

prediction. A similar but more robust mechanism was found in DFI+SPPT (Fig. 18c) and Best Member (Fig. 18d); therefore, they produced better results among all.

For the case of Nivar also, CNTL (Fig. 18e) exhibited relatively weaker ALHF than tuned experiments (Fig. 18f-h). The location of vortex formation and TC's initial movement were quite similar to all the experiments. First, they travelled north-westward along with the actual track and then turned to the left and moved in this

direction for the next few hours before turning to the right side. Due to the weaker magnitude of ALHF over the actual track, the vortex did not cross the observed track, propagated north-westward and made landfall in the south of the actual location. On the other hand, due to relatively strong ALHF from tuned experiments over the actual track, they crossed the observed track. They made landfall close to the actual location, resulting in a lesser track error.

**6 Conclusion**

The performance of a perturbation and filtering-based ensemble technique in the initial and boundary conditions was evaluated using the Advanced Research WRF model with a horizontal resolution of 9 km over the North Indian Ocean (NIO). In this regard, simulations were performed for two tropical cyclones (TCs); (i) Tauktae, which was occurred over the Arabian Sea in May 2021, and (ii) Nivar occurred over the Bay of Bengal during

November 2020. The model results were compared with observations for various aspects of the TCs. Namely, three kinds of experiments with a large number of simulations for both TCs were performed as follows:

(i)   a control experiment (CNTL) in which the initial and boundary conditions were used without any filtering and perturbation.

(ii)   a DFI simulation in which DFI low-pass filter was applied using the scheme available in WRF.

(iii)   DFI+SPPT simulation, which was the mean of ensemble members with perturbation and filtering.

A fixed "Best Member" among all ensembles was considered in the analysis, in addition to the above three sets of experiments. Its selection was based on a comparison of simulated and observed TC trajectories.

The simulated TC's characteristics, such as track and intensity in terms of minimum sea level pressure: MSLP and maximum sustained wind speed: MSW, were demonstrated concerning the available observation of the

India Meteorological Department (IMD) "best-track" dataset. A substantial improvement was noticed in the TC's track simulation with DFI and DFI+SPPT experiments compared to CNTL for both TCs. The along-track (AT) error was improved by 68.8% (65.3%) for Tauktae and 28.2% (37.0%) for Nivar in DFI (DFI+SPPT) compared to CNTL. Similarly, the cross-track (CT) error improved by 23.4% and 40.7% in DFI, 27.7% and 54.1% in DFI+SPPT compared to CNTL for Tauktae and Nivar. The Best Member performed slightly less CT

error compared to DFI and DFI+SPPT. TC intensity, DFI+SPPT outperformed CNTL and DFI by a significant margin. Compared to CNTL, Tauktae has a 38.3% (29.2%) improvement in MSW (MSLP). Similarly, in the case of Nivar, DFI+SPPT showed a 36.2 % (12.7 %) improvement in MSW (MSLP) relative to CNTL.



The spatial patterns of 24 hours accumulated rain during the maximum intensity from all experiments were compared with observations (MSWEP) for both TCs for qualitative demonstration. For Tauktae, an overestimation from CNTL and DFI experiments, whereas DFI+SPPT and Best Member experiments performed reasonably well. In the context of Nivar, all of the simulations were able to capture the rainfall patterns with considerable overestimation. Apart from this, the quantitative assessment was also performed using equitable threat score (ETS) and bias score. In the case of Tauktae, bias scores from all experiments were close for lower thresholds values (50-200 mm/day); however, DFI+SPPT (DFI) experiments showed minimal (maximum) bias score for higher thresholds (200-300 mm/day). Best Member had the best results in ETS, followed by DFI+SPPT and DFI, while CNTL had the least ETS. On the other hand, the bias score and ETS varied slightly across all experiments; nonetheless, DFI+SPPT had the highest ETS for higher threshold values (130-175 mm/day) among all the experiments for Nivar. Overall, DFI+SPPT experiment showed the best results for NIO TCs' simulations.

In brief, the two cases, Tauktae and Nivar, had shown different characteristics in their evolution and dynamics. The Tauktae was formed during the pre-monsoon season (in the month of May) over the ARB, whereas Nivar was developed over the BoB in the post-monsoon season. During their lifespan, Tauktae (Nivar) reached ESCS (VSCS) categories; however, Tauktae showed a rapid intensification event while Nivar did not. The Nivar travelled largely in the northwest direction towards the southeast coast of India, while the propagation of Tauktae was largely northward along the west coast of India. These characteristics of the selected storms were well addressed in the model simulations, and the results were in good agreement with the observations. The SPPT based ensemble mean approach with digital filter initialization in the WRF model has shown considerable improvements in detecting the cyclone characteristics compared to other experiments. This methodology may provide a significant add-on to the real-time operational forecasting of the NIO cyclones.

**Code/Data availability**

The IMD "best-track" data is available at https://rsmcnewdelhi.imd.gov.in/report.php?internal_menu=MzM=. The GFS and FNL data can be accessed via the NCAR Research Data Archive. The MSWEP data is available at http://www.gloh2o.org/mswep/. The ERA5 and MERRA2 reanalysis data can be accessed via the ECMWF climate data centre and NASA Global Modeling and Assimilation office, respectively. The ASCAT satellite product can be taken from http://apdrc.soest.hawaii.edu/data/data.php?discipline_index=3, while the source code of WRF can be downloaded from the https://www2.mmm.ucar.edu/wrf/users/.

**Author contribution**

First and second authors contributed to conceptualization, model simulation, methodology, data analysis, writing the original draft. Third author contributed to conceptualization, resources, supervision, and writing. First, second, third, and fourth authors have contributed to review and editing the manuscript.

**Competing interests**



The authors declare that they have no known competing financial interests or personal relationships that could have appeared to influence the work reported in this manuscript.

### Acknowledgement

440    The first author is thankful to the Department of Science and Technology, Government of India, for giving the DST-INSPIRE research fellowship, registration number IF160165. Indian Institute of Science Education and Research (IISER) Bhopal has provided high-performance computing facilities and a lab environment. Pankaj Kumar acknowledges funding from the Department of Science and Technology (DST), Government of India, grant number DST/INT/RUS/RSF/P-33/G, and the Russian Science Foundation (Project No.: 19-47-02015).

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

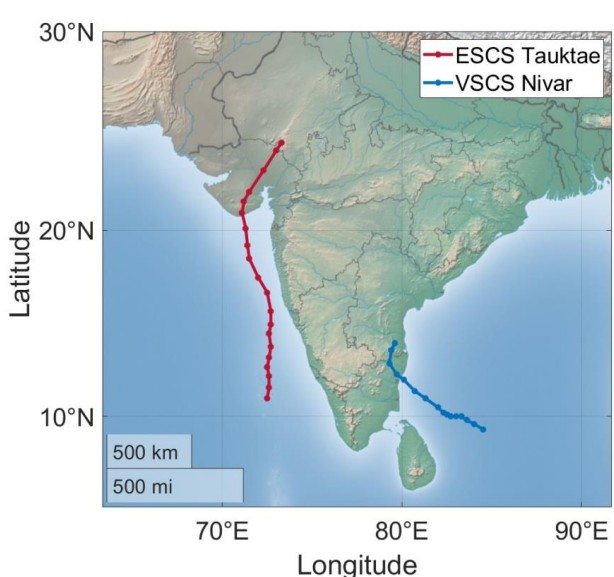

**Figure 1: Observed track of ESCS Tauktae and VSCS Nivar obtained from IMD best-track dataset.**

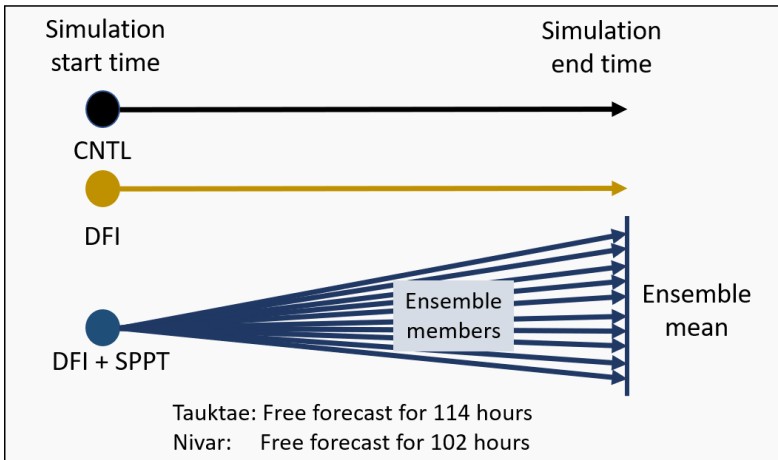

**Figure 2: Graphical representation of experimental setup. CNTL denotes the control simulation, DFI denotes the simulation with DFI implementation on CNTL, and DFI+SPPT depicts the SPPT ensemble simulation's mean with DFI.**



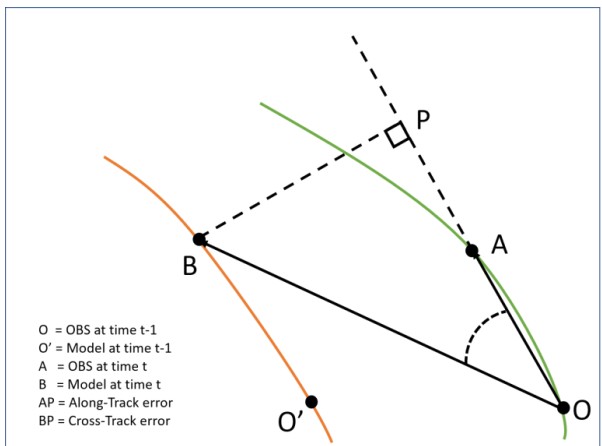

**Figure 3: Schematic diagram representing the mechanism for calculating CT and AT error.**

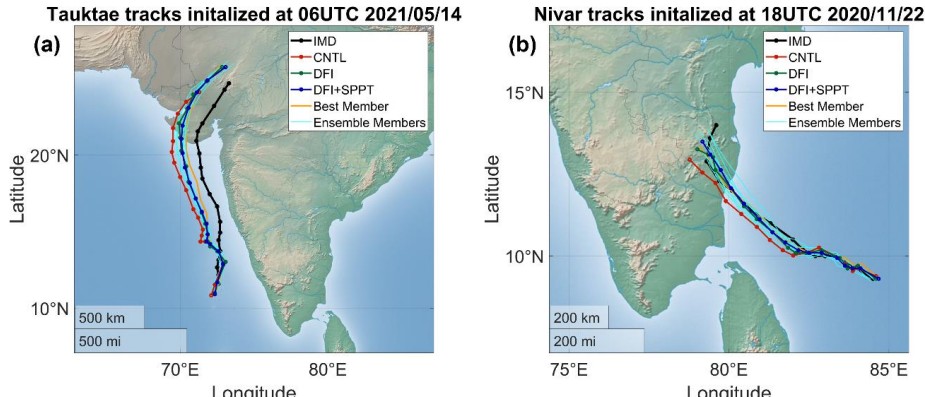

**Figure 4: Observed and simulated tracks from all the experiments (CNTL, DFI, DFI+SPPT) for TCs a) Tauktae,**

**and b) Nivar.**

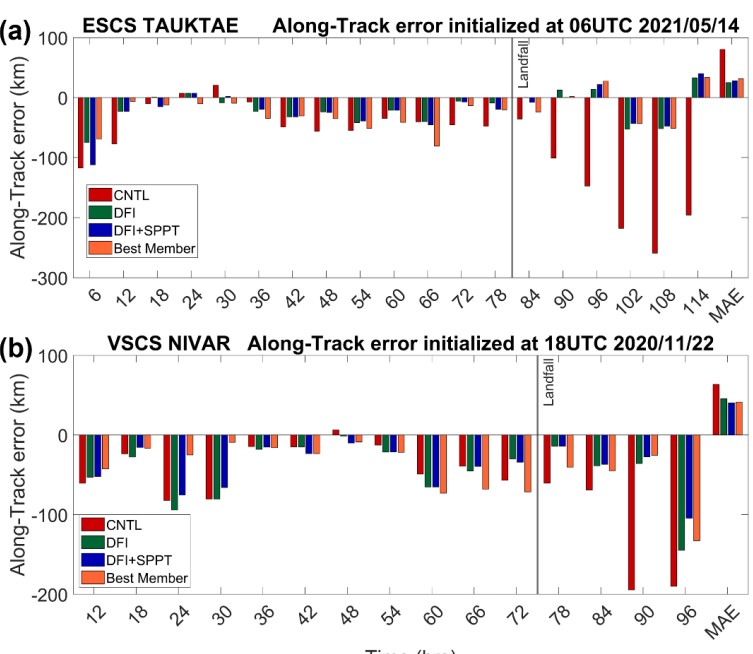

**Figure 5: AT error bars at every 6-hour interval against the IMD best-track data for two TCs, a) Tauktae, b) Nivar.**

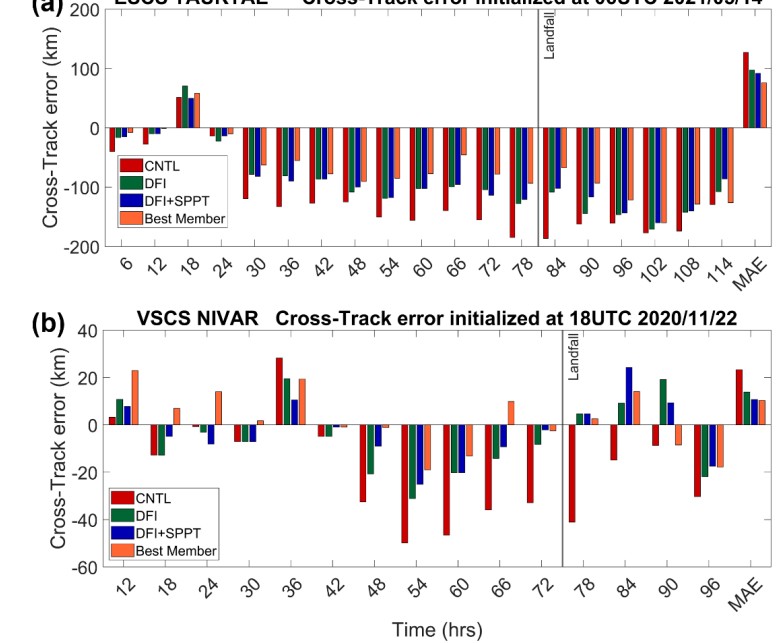

**Figure 6: Same as Figure 5 but for CT.**






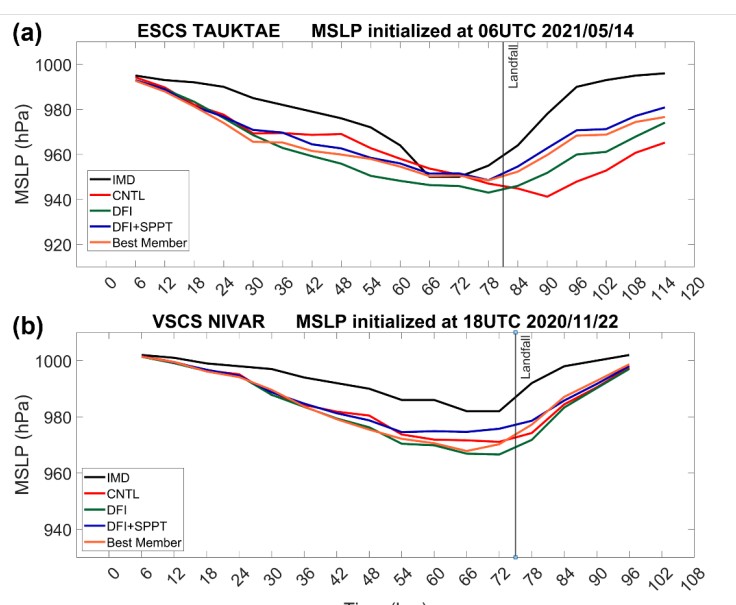

**Figure 7: 6-hourly time evolution of MSLP for TCs a) Tauktae, and b) Nivar from observation and model experiments.**

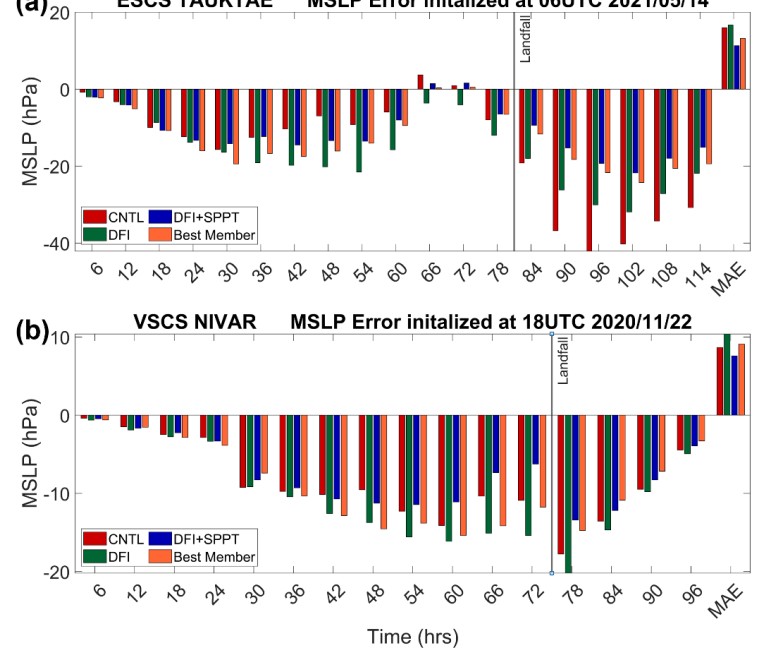

**Figure 8: Error in MSLP prediction against the IMD best-track data for TCs a) Tauktae and b) Nivar.**





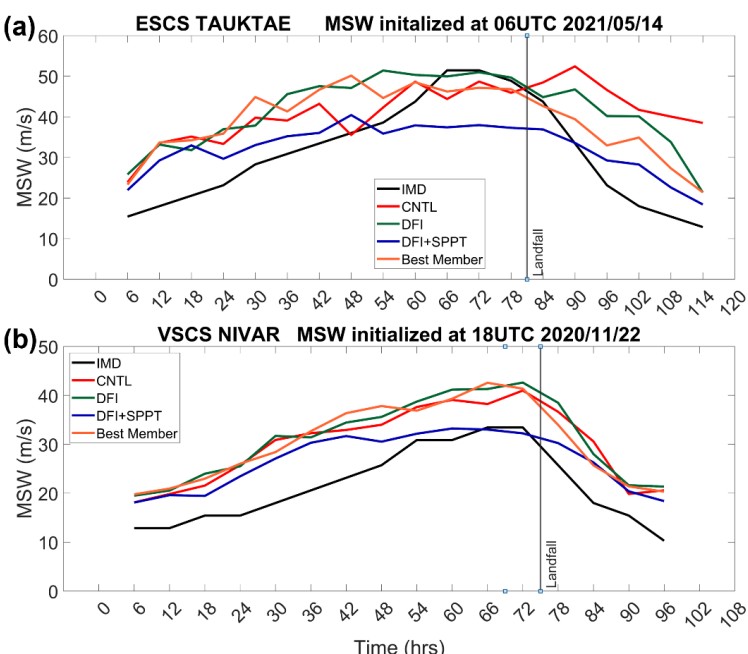

**Figure 9: Same as Figure 7 but for MSW.**

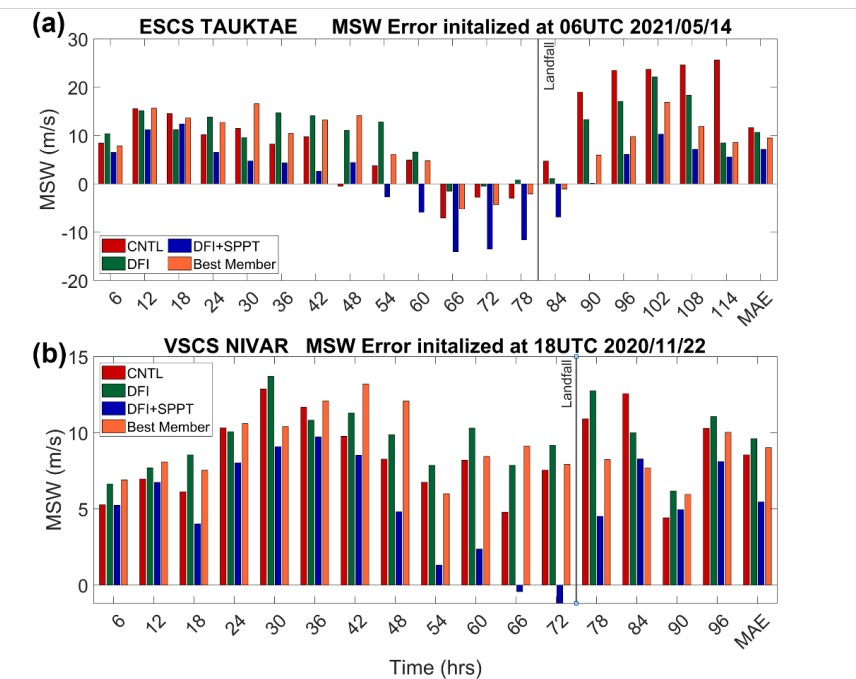

**Figure 10: Same as Figure 8 but for MSW.**





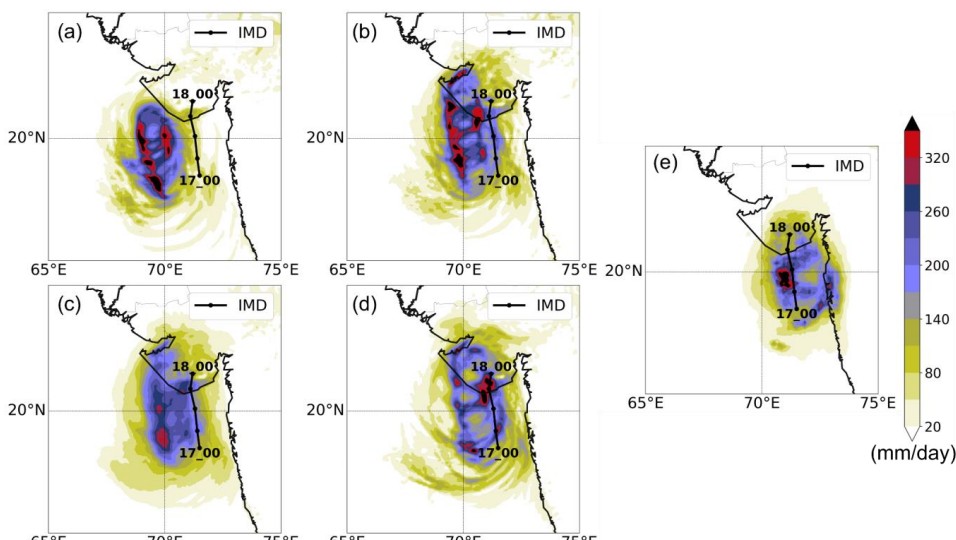

**Figure 11: Comparison of 24-accumulated rainfall (mm) for ESCS Tauktae from a) CNTL experiment, b) DFI experiment, c) DFI+SPPT experiment, d) Best Member, and e) MSWEP observation.**


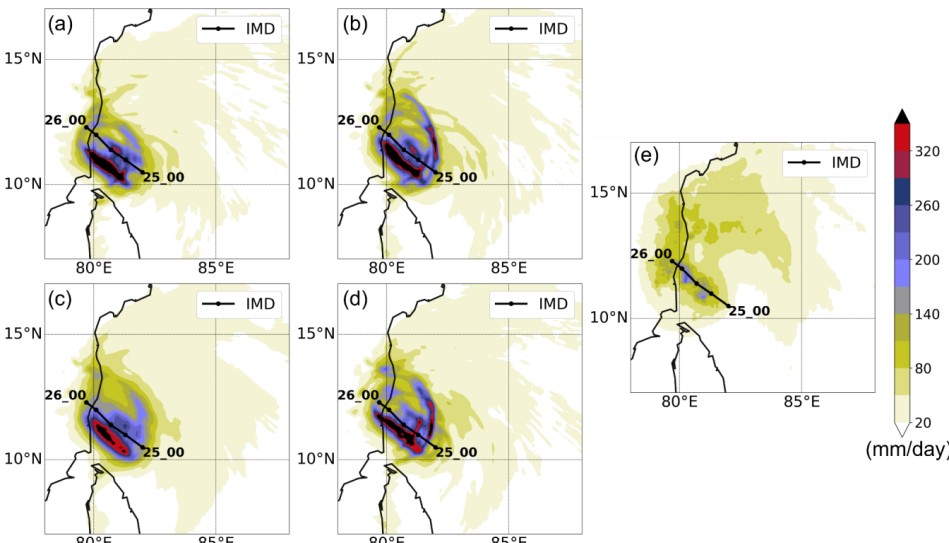

**Figure 12: Comparison of 24-accumulated rainfall (mm) for VSCS Nivar from a) CNTL experiment, b) DFI experiment, c) DFI+SPPT experiment, d) Best Member, and e) MSWEP observation.**




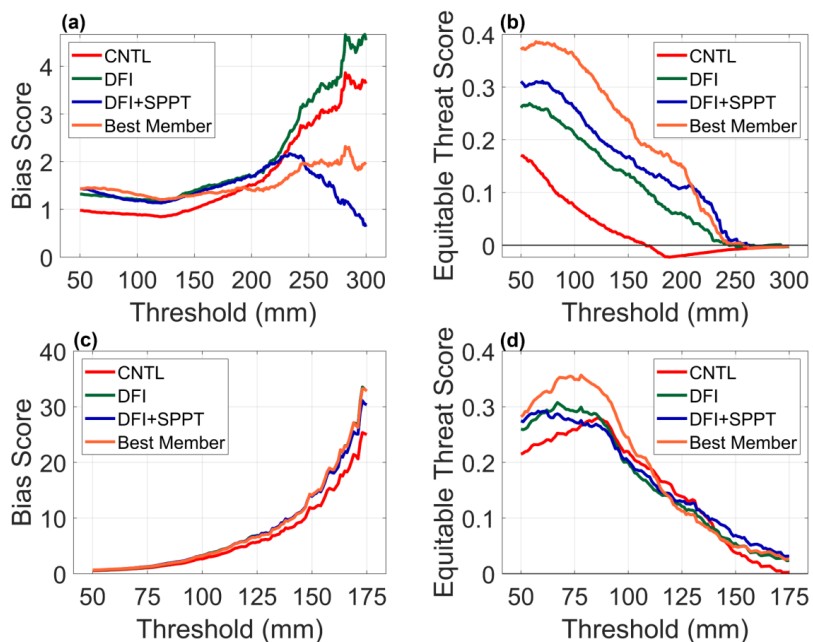

**Figure 13: Bias score (left column) and equitable threat score (right column) in the 24-hour accumulated rainfall prediction from the model experiments at the various threshold for two cases Tauktae (00 UTC 17 May - 00 UTC 18 May 2021; first row) and Nivar (00 UTC 25 November - 00 UTC 26 November 2020; second row). The calculation is performed for a domain of 10º × 10º extended from 65ºE to 75ºE, 15ºN to 25ºN and 78ºE to 88ºE, 7ºN to 17ºN for Tauktae and Nivar, respectively.**

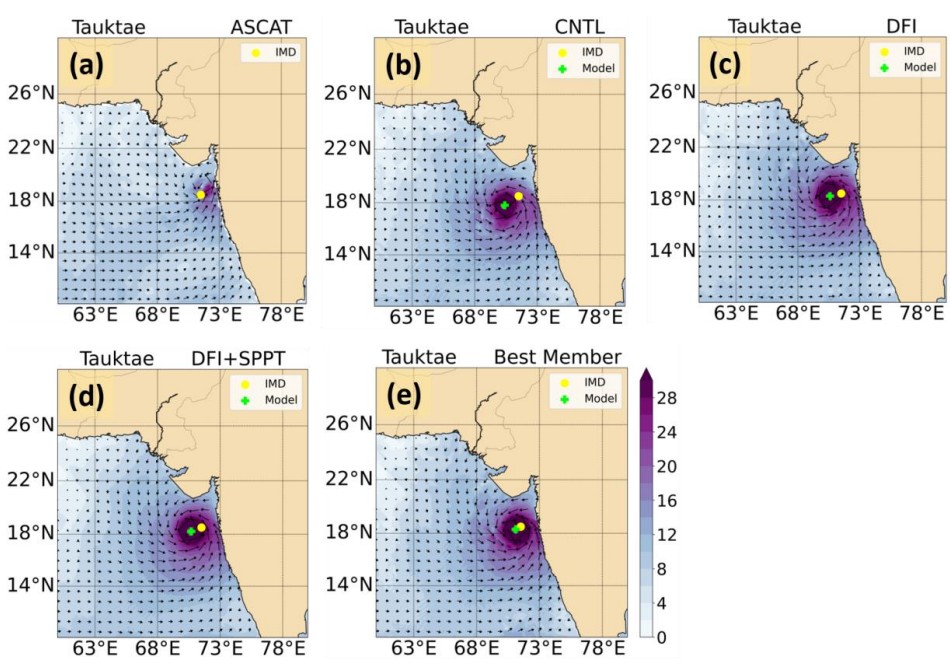



**Figure 14: Horizontal wind distribution at 10-meter height over the ocean valid at 2021-05-17_00 for cyclone Tauktae from a) observation (ASCAT); b) CNTL; c) DFI; d) DFI+SPPT; and e) Best Member experiments.**

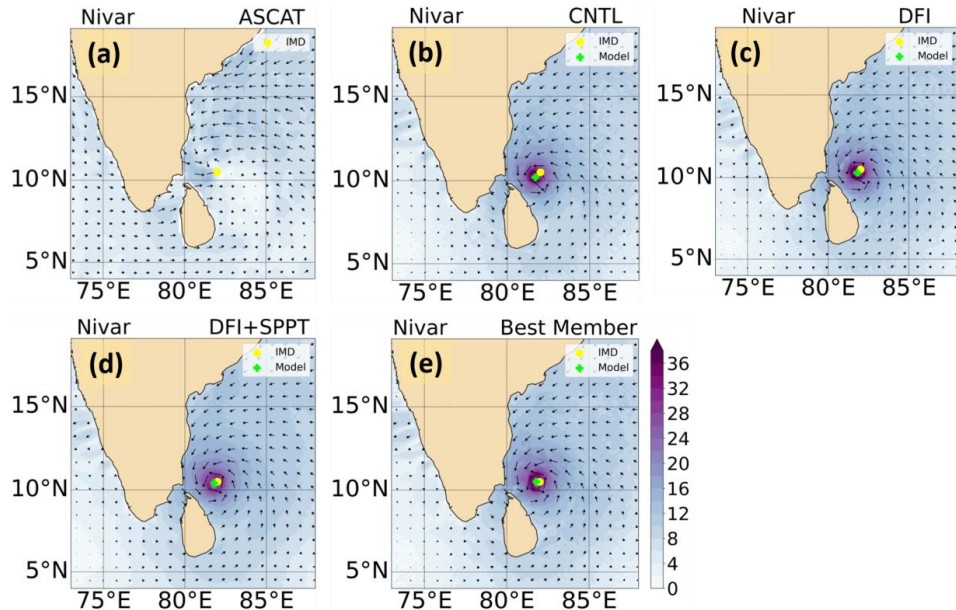

**Figure 15: Horizontal wind distribution at 10-meter height over the ocean valid at 2020-11-25_00 for cyclone Nivar from a) observation (ASCAT); b) CNTL; c) DFI; d) DFI+SPPT; and e) Best Member experiments.**



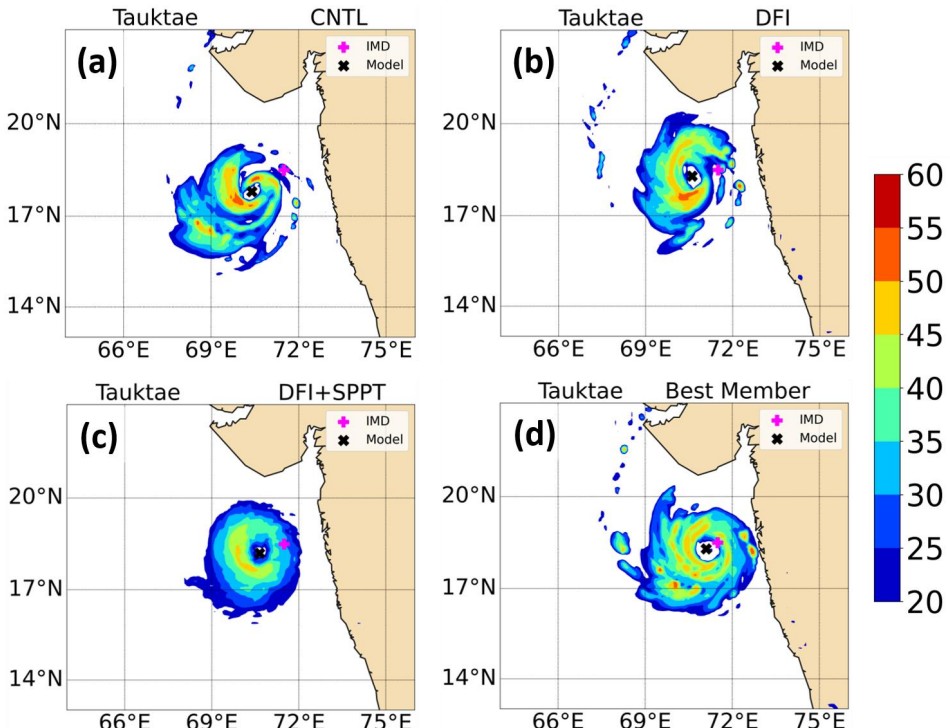

**Figure 16: Spatial distribution of the maximum reflectivity of cloud (unit: dBZ) during the maximum intensity time valid at 2021-05-17_00 UTC for cyclone Tauktae from a) CNTL; b) DFI; c) DFI+SPPT; and d) Best Member experiments.**


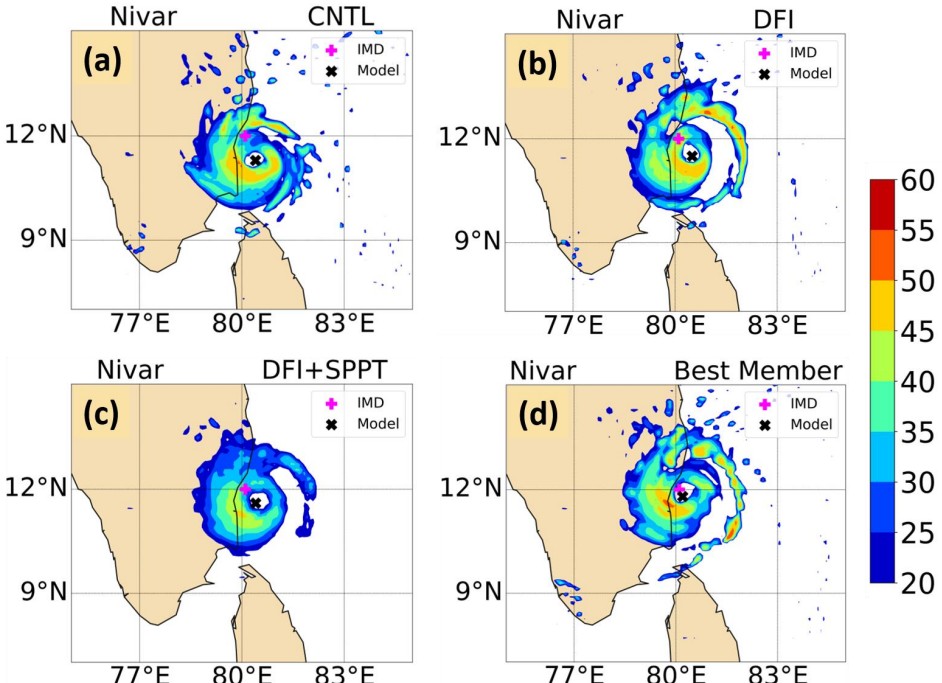

**Figure 17: Spatial distribution of the maximum reflectivity of cloud (unit: dBZ) during the maximum intensity time valid at 2020-11-25_18 UTC for cyclone Nivar from a) CNTL; b) DFI; c) DFI+SPPT; and d) Best Member experiments.**

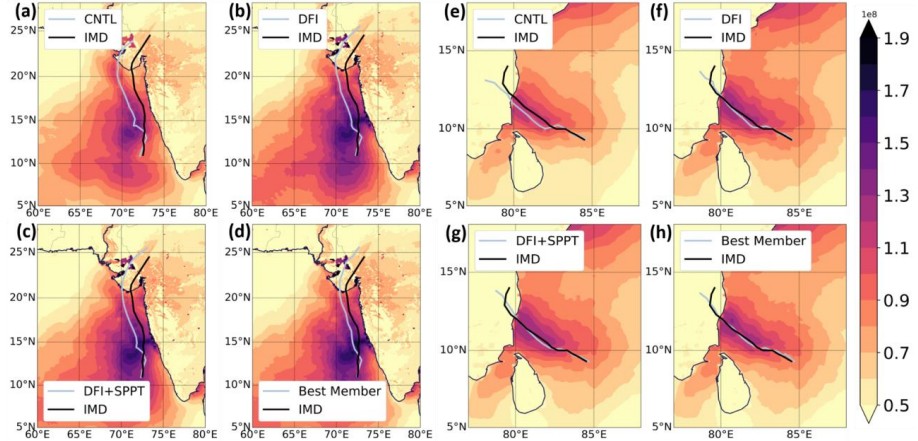

**Figure 18: Model simulated accumulated upward latent heat flux at the surface (J/m²) for TC Tauktae (a-d) and Nivar (e-h). Colorbar values are the multiple of 1×10⁸. The TC's track from the observation and respective experiments are also plotted.**

| Parameter | Value | Description |
|---|---|---|
|  |  |  |



| gridpt_stddev_sppt | 0.2 | The standard deviation of the random perturbation field at each grid point |
| stddev_cutoff_sppt | 2.5 | Cut-off tails of perturbation pattern above this threshold standard deviation |
| lengthscale_sppt | 150,000 meters | Random perturbation length-scale |
| timescale_sppt | 21,600 seconds | Temporal decorrelation of random field |

**Table 1: Description of SPPT parameters used in this study.**
