# Peer review of "Stochastically perturbed physics-tendencies based ensemble mean approach in the WRF model: a study for the North Indian Ocean tropical cyclones"

_Weather and Climate Dynamics, 2022_

## Author Comment (AC1)

Response for the manuscript "Stochastically perturbed physics-tendencies based ensemble mean approach in the WRF model: a study for the North Indian Ocean tropical cyclones" by Gaurav Tiwari et al., Weather Clim. Dynam. Discuss., https://doi.org/10.5194/wcd-2022-10-RC1, 2022

We thank the editor and the Anonymous Referee #1 for their valuable comments. The suggestions have certainly improved the manuscript quality. A pointwise reply to the reviewer's comments is provided below.

**Anonymous Referee's Comments**

**General assessment**

The authors analyse the ensemble forecast performance of tropical cyclones when a stochastic disturbance is added to the physics trends (SPPT) in the WRF model. They also add a numerical filter initialization (DFI) to the initial state of the forecast. The performance of the DFI+SPPT ensemble forecast is evaluated against a control simulation and against a simulation that uses only DFI. These various model set ups are use to forcast two tropical cyclones which developed in the North Indian Ocean. The authors show that for a number of parameters evaluating the transverse and longitudinal position error, intensity, precipitation associated with CT, DFI+SPPT performs better. Therefore, they recommend that it be used in future forecasts made with WRF.

The paper and especially the result section describe the figures (which are of good quality) in great detail: there are many numbers and acronyms which sometimes make the reading difficult. The text could sometimes be simplified. While the conclusion merely summarises the main results, what is striking is the absence of any discussion of the results in the paper, which I think is necessary.

From a scientific point of view, the study of the impact of the SPPT parameterisation is relevant, since its usefulness for forecasting is debated. On the other hand, the evaluation made here is questionable: the DFI+SSPT ensemble is compared to a deterministic forecast of the CNTL and DFI, which should not allow to conclude (I develop this concern below). The difference between DFI+SPPT and the other experiments is always interpreted as due to the introduction of SPPT but could be equally due the size of the ensemble. Similarly, the difference between CNTL and the other experiments is always interpreted as the introduction of DFI, but could be equally due to the retuning that has been performed.

I therefore recommend that the paper be reconsidered for publication after a major revision.

**Major concerns**

**Comment 1:** I have a major concern about the comparison between DFI and DFI+SPPT. This comparison is essential because it supports the only major conclusion of the paper : "The SPPT based ensemble mean approach with digital filter initialization in the WRF model has shown considerable improvements in detecting the cyclone characteristics compared to other experiments."

The issue is that DFI has one single member while DFI+SPPT is an ensemble of 10 members. The reduction of error in DFI+SPPT could result from a better sampling of possible outcomes. A possible evidence of that is that the "best member" of DFI+SPPT has a comparable score to

DFI for the intensity metrics and is often in lesser agreement with the ensemble mean of DFI+SPPT, although it is one of its member!

A more rigorous assessment should compare two ensembles of similar sizes for DFI and DFI+SPPT.

**Reply:** Thank you for the comment. The essential objective of the study is to elaborate on the impact of a digital filter initialization (DFI) process within the WRF model on the historical prediction (simulation) of tropical cyclone activities over the North Indian Ocean. With this approach, we found remarkable improvements in the cyclone's track simulation (compared to the CNTL) but very less impact of DFI on the cyclone's intensity simulation. Thus, this prototype setup can be used for further future studies.

Moreover, to improve the skill of the WRF model (with DFI) for cyclone's intensity simulation, we generated the ensemble members using SPPT. We agree with the reviewer, that the reduction of error in DFI+SPPT could result from a better sampling of possible outcomes. Various studies have shown the improved precipitation in the ensemble mean (even further improved performance with increasing ensemble members) by comparing the control simulation with the ensemble mean of the perturbed simulation.

This study is not focused on the source of improvement but rather on obtaining the possible best setup and the results indicate the reduction of error in DFI+SPPT.

It is possible that in the large ensemble, some of the ensembles might be worse than in control which might lead to worsening the performance of the ensemble mean. In line with this, the purpose of keeping the Best Member was to illustrate the behaviour of a single member also, including the mean of all the members. Since the selection of the Best Member was based on cyclone's track simulation only, it exhibited a comparable score to DFI for the intensity metrics.

**Comment 2:** There is no mention of an ocean model in the model setup, so I assume that all experiments are atmosphere-only. It seems necessary to describe the SST product. In particular, because the surface latent heat flux is argued to be the cause of the errors in track of CNTL.

**Reply:** We thank the reviewer for the suggestion. This study uses a standalone atmospheric model (WRF) for all the experiments. We used a fixed SST value for the simulations, which is a necessary criterion for the WRF simulations with active DFI mode. Therefore, the SST product is not described in the manuscript. Having been an atmosphere model, the feedback of the oceanic processes is not attributed to the atmospheric processes; however, with a fixed SST also, the WRF model provides the value of surface latent heat flux as a variable at the user-specific time interval.

**Comment 3:** It is likely that there is no or a strong underestimation of the cold wake feedback, and as such, it is not surprising that the experiments tend to overestimate the intensity of the two TCs. The fact that DFI+SSPT captures the peak intensity of Nivar, while the other experiments overestimate it, is probably not a good thing, as the cyclone would have been weaker with a SST cooling. The authors should discuss that.

**Reply:** Thank you. We agree with the reviewer that underestimating the cold wake feedback in the model simulations can overestimate the intensity of the simulated cyclones. However, better sampling of the ensemble members in the DFI+SPPT experiments is the plausible reason for capturing the peak intensity of the cyclones regarding the observation.

In this study, the SST was kept constant throughout the simulation length (because of the requirement for the DFI scheme activation). Due to this, other mechanisms could probably be responsible for a such overestimation in the cyclone's intensity, especially from the CNTL, DFI, and Best Member experiments, which might be a great motivation for further research.

**Comment 4:** In their analysis of Fig. 11 and Fig. 12, the authors analyse the fact that the precipitation intensity is reduced in DFI+SPPT and in closer agreement with the observations as an improvement due the SPPT scheme. But it is most likely the result of averaging the ensemble. Again, an evidence of that is that the best member of DFI+SPPT has more intense precipitation than DFI+SPPT ensemble mean.

**Reply:** Thank you for the comment. We agree with the reviewer that improvement in the SPPT scheme is likely the result of averaging the ensemble. We included this discussion in the revised manuscript in Line No. 306-310 as follows:

"Thus, In Fig. 11 and Fig. 12, the cyclone-induced rainfall was better simulated by the DFI+SPPT experiment than by CNTL and DFI, mainly due to the better sampling of the ensemble members generated by the SPPT scheme. It illustrates that a set of small ensembles can enhance the WRF model framework skill for rainfall prediction/simulation."

**Comment 5:** A retuning of DFI and DFI+SPPT has been performed. Which parameters have been retuned? This retuning is as likely to explain the differences in track between CNTL and the other experiments, as the introduction of a DFI in the initial state. It should be described.

**Reply:** Thank you for this important comment. The setting for SPPT parameters is provided in Table 1. The standard deviation value of the random perturbation field at each grid point was set to 0.2. Random perturbation length-scale and temporal decorrelation of the random field were set to 150,000 meters and 21,600 seconds, respectively.

The (tuning or the) setting of the DFI parameters was based on the literature survey.

The tuning details and related reference is included in the revised manuscript as follows:

"A filter based on the Dolph-Chebyshev window (Lynch, 1997) was used in this study. The filter's backstop was set to 60 minutes, while the forward stop was set to 30 minutes".

*Lynch, P.: The Dolph–Chebyshev window: A simple optimal filter. Monthly weather review 125, 655-660, 1997.*

**Comment 6:** The authors suggest on the contrary that the lesser surface turbulent heat flux is the cause of the difference in CNTL track for Tauktae: could they test their hypothesis? I believe that the different sets of parameters would cause CNTL to track more west than the retuned experiments, which would cause lesser surface turbulent heat flux, rather than the contrary.

**Reply:** Thank you for the comment. Regarding mentioned point, it was found from the analysis that the higher (lesser) accumulated upward latent heat flux in the left (right) of the simulated track was one of the primary causes of the difference in the CNTL track for Tauktae. As suggested, further we tested our argument by evaluating the simulated spatial distribution of lower tropospheric (950 hPa) absolute vorticity for cyclone Tauktae from all the experiments (shown in Fig. S12). We selected two successive time steps ($4^{th}$ and $5^{th}$) to capture the behaviour of the cyclonic vortex toward its movement.

Particularly from the CNTL simulation, the contour of vorticity maxima at the $4^{th}$ time step (Fig. S12a) was in the west of the cyclone's track which got shifted to the south of the track in

the 5th time step (Fig. S12b); whereas, the storm still travelled north-westward. More or less, a similar sort of behaviour was seen in other experiments too. Thus, it can be said that upward latent heat flux played a pivotal role in directing the cyclone Tauktae from CNTL simulation more west than the retuned experiments.

[Figure]

**Figure S12: Spatial distribution of absolute vorticity of cyclone Tauktae in the lower troposphere (950 hPa) plotted for the 4th (first row) and 5th time steps (second row) for CNTL (first column), DFI (second column), DFI+SPPT (third column), and Best Member (fourth column) experiments.**

**Minor revisions**

**Comment 7:** line 33 : "frequent and intense TCs" is a confusing statement. Does it mean "more frequent and more intense TCs" : there is certainly no consensus on an increase of TC frequency! Or do they mean "more frequent intense TCs"?

**Reply:** Thank you for the comment. In this statement, we want to convey that the frequency and intensity of TCs in the ARB is expected to increase compared to the historical database. The sentence has been rephrased in the revised manuscript as follows:

In the NIO, particularly in the ARB, the frequency of TCs is expected to increase compared to the historical database. At the same time, their intensity will also increase as the earth continues to warm (Deshpande et al., 2021).

**Technical issues**

**Comment 8:** line 52 : influence -> influences

**Reply:** Thank you. The suggestion has implemented in the revised manuscript, Line no. 58.

**Comment 9:** line 125 : Why were two different convection schemes used?

**Reply:** Thank you for the comment. The convection scheme is one of the key constraints that control the model performance. Recent studies found that the same scheme is performing best for one region while worse for other regions and suggested using the mixed convection scheme

(different scheme over the different areas) (Mishra and Dwivedi et al., 2019). With these lines, We performed some sensitivity runs and selected the best-performing convection scheme. The Kain-Fritsch cumulus convection scheme over the ARB and Grell 3D Ensemble scheme over the BoB domain to get the best-simulated tracks of TCs. Rest other schemes were similar for both domains.

**Comment 10:** line 198 : from the Best Member -> for the Best Member.

**Reply:** Thank you. The suggestion has implemented in the revised manuscript, Line no. 205-206.

**Comment 11:** line 384 : which was occured -> which occurred

**Reply:** Thank you. The suggestion has implemented in the revised manuscript, Line no. 414.

---

## Author Comment (AC2)

Response for the manuscript "Stochastically perturbed physics-tendencies based ensemble mean approach in the WRF model: a study for the North Indian Ocean tropical cyclones" by Gaurav Tiwari et al., Weather Clim. Dynam. Discuss., https://doi.org/10.5194/wcd-2022-10-RC2, 2022

We thank the editor and Anonymous Referee #2 for their valuable comments. The suggestions have certainly improved the manuscript quality. A pointwise reply to the reviewer's comments is provided below.

Anonymous Referee's Comments

The paper mainly describes results from WRF ensemble model simulations at 9 km grid spacing using stochastically perturbed physics-tendencies. The authors use the ensemble mean approach along with digital filter initialization to the initial and boundary conditions to analyse two TCs that made landfall in India. The experiments showed that the ensemble-mean approach, the digital filter initialization approach, and the combination of both methods lead to improvements in the track forecast. Only the combined methods showed an improvement in intensity forecast.

**Major comments**

**Comment 1:** I'm struggling to see how the paper explains processes (dynamics) of TCs and what we learn about TCs.

**Reply:** Thank you for raising the concern that helped us to clarify the text and improve the manuscript. For the sake of clarity, we added a section in the revised manuscript describing the process (dynamics) of the TCs as follows:

**"5.3 Cyclones' vertical distribution**

The pressure-longitude vertical distribution of absolute vorticity ($10^{-5}$ s$^{-1}$) valid at the time of highest MSW speed of cyclones Tauktae and Nivar from the four experiments (CNTL, DFI, DFI+SPPT, and Best Member) are illustrated in Fig. 19 and 20, respectively. The latitude was averaged between 3$^o$ in the north and 3$^o$ in the south from the cyclones' central location corresponding to the individual experiment.

In the case of Tauktae, all the experiments show high positive absolute vorticity (decreasing from lower to upper troposphere) around the eye of the cyclone with a slight difference in term magnitude and vertical extent zonal extent (Fig. 19). The CNTL and DFI simulations show the comparatively higher vertical extension of high absolute vorticity (40-50$\times10^{-5}$ s$^{-1}$) up to 400 hPa from the surface than DFI+SPPT and Best Member simulations in which this maximum vorticity zone ix restricted only up to 600 hPa. The vorticity strength in the vertical column is consistent with the distribution of the highest MSW speed shown in Fig. 9a.

A similar inference for the vertical distribution of absolute vorticity is noticed for the cyclone Nivar (Fig. 20) has played a crucial part in addressing the intensity of cyclone Nivar also slightly different from Tauktae. The overall strength of the vorticity was found to be weaker by one to two orders (approximately 10-20$\times10^{-5}$ s$^{-1}$) which matches the categories of these two TCs. However, the height of the vorticity in the column around the cyclone's centre was above 500 hPa in all experiments. The DFI experiment attained the maximum height (approximately 350 hPa), followed by Best Member, CNTL, and DFI+SPPT experiments in decreasing order. Also, the vorticity maxima from the DFI and Best Member experiments were in the order of 40-50$\times10^{-5}$ s$^{-1}$ in the lower atmosphere, which was larger than in the CNTL and

DFI+SPPT experiments (30-40×10$^{-5}$ s$^{-1}$). This distribution of the vorticity strength in the vertical column is again consistent with the distribution of the highest MSW speed of cyclone Nivar, as shown in Fig. 9b.

[Figure]

**Figure 19: Pressure-longitude vertical distribution of absolute vorticity (unit: 10$^{-5}$ s$^{-1}$) extended from lower troposphere (900 hPa) to upper troposphere (100 hPa) for cyclone Tauktae computed at the time of peak MSW from a) CNTL, b) DFI, c) DFI+SPPT, and d) Best Member experiments. Here, a 6° average of the latitude was considered, i.e. extended to 3° in the north and 3° in the south from the storm's central location (corresponding to the concerned experiment).**

[Figure]

**Figure 20: Same as Figure 19, but for cyclone Nivar.**

**Comment 2:** From reading the introduction the main aim of the paper is to compare different WRF model configurations and their ability to forecast TCs. I can see the rationale behind that, but the model experiments do not give us new insight into understanding TCs. Maybe the benefits and links to operational forecasting could be made clearer.

**Reply:** Thank you for the comment. The better the cyclone prediction is vital for the socioeconomic perspective. The improved initial condition is crucial for improving cyclone prediction. In this regard, the study's primary objective is to elaborate on the impact of a digital filter initialization (DFI) process within the WRF model on the simulation of tropical cyclone activities over the North Indian Ocean. With this approach, we found remarkable improvements in the cyclone's track simulation (compared to the CNTL) but very less impact of DFI on the cyclone's intensity simulation. Moreover, to improve the skill of the WRF model (with DFI) for cyclone's intensity simulation, we generated the ensemble members using SPPT. Thus, we have compared all the experiments, i.e., CNTL, DFI, and DFI+SPPT. Ultimately the DFI+SPPT configuration provided the more realistic properties of the TCs. In a further study, we are exploring new insight into understanding TCs which is not the objective of this manuscript.

**Comment 3:** At 9-km grid spacing convection has to be parameterised. Several studies have shown that going to higher resolutions of 4 km and below, where the convective parametrisation needs to be switched off, the intensity of TCs is captured better than in coarser model runs. Have you tested the impact of grid spacing on the model results?

**Reply:** Thank you for the valuable comment. This study used the convection parameterization subjected to the 9-km grid spacing. We agree with the reviewer that going to higher resolutions of 4 km and below (i.e. convection-permitting resolution), the convection is explicitly resolved and no need to be parameterized. In this study, we have not tested the impact of grid spacing (convection-permitting resolution) on the model results; however, in the near future, we will perform a comprehensive analysis on these lines over the same study region using the best setup obtained in this study.

**Comment 4:** Are you using fixed SST values for the simulations?

**Reply:** We have used a fixed SST value for the simulations, which is a necessary criterion for the WRF simulations with active DFI mode.

**Comment 5:** The ensembles seem not to have a lot of spread, even when you are randomly perturbing the initial conditions and perturbing the stochastic physics. Might your ensemble be underrepresenting the uncertainties?

**Reply:** Thank you for the comment. We agree that the ensembles (particularly in cyclone Tauktae) do not seem to have much spread. One reason for this could be the size of the ensembles (which is ten). However, the spread is relatively wider for cyclone Nivar. Thus, we believe that ensembles have reasonably represented the uncertainties.

**Comment 6:** Do you have any observations (maybe microwave satellite data/imagery) that gives insights into the observed structure of both storms? I like the plots that show rainfall accumulation in comparison to the observed rainfall. How good is ASCAT at representing the windspeed in TCs?

**Reply:** Thank you for the comment. We used the ASCAT wind product in this study and have the scarcity of any observation source. The wind products of the ASCAT instrument aboard the European Organization for the Exploitation of Meteorological Satellites (EUMETSAT)

Metop-A are processed by NOAA/National Environmental Satellite, Data, and Information Service (NESDIS). The radar sensor measures backscatter to calculate the speed and direction of winds over the ocean's surface. ASCAT data is used to feed numerical weather prediction models and gives significant information about storm activity.

**Comment 7:** I find that parts of the paper are quite descriptive.

**Reply:** We agree with the reviewer that the mentioned parts of the manuscript are slightly descriptive but reducing the text count affects the information coming from the figures. Therefore, this portion is marginally descriptive to fulfill the purpose of the study.

**Minor comments**

**Comment 8:** The manuscript (including the abstract) contains too many acronyms, which reduces its readability.

**Reply:** Thank you. The suggestion has incorporated in the revised manuscript wherever it could be possible to make it.

**Comment 9:** 49-50: The text could be worded better. I noticed also in other places that the text could be more precise. I suggest going through the text carefully again.

**Reply:** Thank you. The suggestion has been applied in the revised manuscript as given (Line No. 48-54) and wherever it could be possible to make it.

"The convection scheme is one of the key constraints that control the model performance. Recent studies found that the same convection parameterization scheme is performing best for one region while worse for other regions and suggested using the mixed convection scheme (different scheme over the different areas) (Mishra and Dwivedi et al., 2019). Different physics and parameterization scheme combinations can generate different model results (Mishra and Dubey, 2021. Furthermore, model resolutions are important in predicting skills. The high-resolution helps resolve the complex topography and sub-grid processes (Mishra et al., 2021)."

**Comment 10:** 67, L. 73: Remove "very".

**Reply:** Thank you. The suggestion has implemented in the revised manuscript, Line no. 73.

**Comment 11:** 122: "Horizontal grid spacing of 5 minutes" – It seems something has gone wrong here.

**Reply:** Thank you. Since $1^o = 60$ minutes $= \sim 111$ km in the tropics. Thus, here horizontal grid spacing of 5 minutes are referring a distance of approximately 9 km.

**Comment 12:** 143: Dot instead of colon after Table 1.

**Reply:** Thank you. The suggestion has been implemented in the revised manuscript, Line no. 149.

**Comment 13:** "IMD" is undefined. The definition only comes in l. 395, which is in the conclusions.

**Reply:** Thank you. The suggestion has been implemented in the revised manuscript, Line no. 91.

**Comment 14:** Is the "best member" defined only based on the track?

**Reply:** Yes, the "best member" is defined based on the track.

**Comment 15:** 185: "relatively some more error" – Please be more specific.

**Reply:** Thank you. The sentence has been rephrased in the revised manuscript as follows:

"The DFI was detected with the least error near the landfall (at the 78-hour mark), followed by relatively some more error 19.5 and 20.3 km from the DFI+SPPT and Best Member, respectively."

**Comment 16:** 234: What does "making the results more perspective" mean?

**Reply:** The sentence has been rephrased in the revised manuscript Line No. 241-242 as follows:

"To make the result more robust, we demonstrated the comparative performance of all experiments by computing the error/bias (observation minus model) in MSLP (Fig. 8)."

**Comment 17:** It seems nothing from Section 5 has made it into the abstract. What are the key results from this section?

**Reply:** Thank you for the comment. The key results from Section 5 illustrate that the accumulated upward latent heat flux reasonably controlled the track of cyclones Tauktae and Nivar, whereas the vertical distribution of absolute vorticity had a noticeable impact on the maximum intensity of cyclones. As suggests, the abstract in the revised manuscript has been modified as below:

"Tropical cyclones (TCs) are among the catastrophic natural hazards over the North Indian Ocean (NIO), and they are expected to become more frequent in the upcoming years. TCs occur primarily in the pre-monsoon (April-June) and post-monsoon (October-December) seasons, wreaking havoc on South Asian regions. For reliable alerts and disaster warnings ahead of time, better forecasting of TC features such as track, landfall, intensity, rainfall, and so on is crucial. The present study uses the stochastically perturbed physics-tendencies (SPPT) ensemble-mean approach along with digital filter initialization (DFI) to the initial and boundary conditions for the high-resolution Weather Research and Forecasting (WRF) model. The model's sensitivity has been investigated for the two NIO TCs, Tauktae (in May 2021) and Nivar (in November 2020), by performing a large number of experiments. Compared with control runs, the track simulations in terms of the reduction in along-track (cross-track) errors for Tauktae and Nivar were improved by 68.8% (23.4%) and 28.2% (40.7%), respectively, in the DFI experiment. Further improvements were found in the SPPT-based ensemble mean experiments (DFI+SPPT) as the along-track (cross-track) errors, compared to control simulations, were reduced by 65.3% (27.7%) and 37% (54.1%), for Tauktae and Nivar, respectively. However, the DFI simulations showed a potential to improve the cyclones' track simulation but failed to reduce the error in intensity simulation. On the other hand, DFI+SPPT experiments improved the model's reliability in simulating TCs' intensity (maximum sustained wind speed and minimum sea-level pressure) considerably. Thus, the DFI+SPPT experiments showed higher skills in simulating the cyclones' characteristics. The accumulated upward latent heat flux reasonably controlled the track of cyclones Tauktae and Nivar, whereas the vertical distribution of absolute vorticity had a noticeable impact on the maximum intensity of cyclones."

**Comment 18:** 362: "the tuned experiments" – Are you referring to all the other runs discussed in the manuscript?

**Reply:** Thank you for the comment. Here tuned experiments are all the experiments other than CNTL run.

**Comment 19:** 363-364: "a hot spot in the central-east ARB pulled the cyclonic circulation towards it"

– Not sure what you mean. Are you referring to a maximum upward latent heat flux? The whole section 5.2 could be written in a more precise manner.

**Reply:** Thank you for the comment. Yes, the hot spot refers to a maximum upward latent heat flux. As suggested, the whole section 5.2 has been rewritten in revised manuscript as follows:

"The accumulated upward latent heat flux at the surface level (ACLHF; $J/m^2$) from all experiments explains the propagation of TCs (Fig. 18a-h). In the instance of Tauktae, the initial vortex from CNTL (Fig. 18a) formed southwest of the observed location, whereas the start point of the TCs in the tuned experiments was quite near to the observation (Fig. 18b-d). Again, as shown in Fig. 18a, a hot patch (maximal upward latent heat flux) in the central-east ARB drew the cyclonic circulation towards it. The TC followed the westward passage for a few hours before turning north-eastward and slightly north-westward. This westward migration resulted in a significant error in the TC Tauktae's CNTL experiment track forecast. The magnitude of ALHF was greater than CNTL in the DFI experiment (Fig. 18b), and we can identify two hotspots, one in the southwest and one on the east side of the track. This dipole type distribution ultimately restricted Tauktae's westward progression; as a result, it did not deviate much from the actual trajectory, resulting in a reduced error in track prediction. DFI+SPPT (Fig. 18c) and Best Member (Fig. 18d) both had a similar but more robust mechanism; therefore, they generated better results among all.

In the case of Nivar, CNTL (Fig. 18e) revealed weaker ALHF than tuned experiments (Fig. 18f-h). The location of vortex formation and the initial movement of TC were very similar across all experiments. They started by moving north-west along the actual track, then veered to the left and continued for a few hours until turning to the right. The vortex did not intersect the observed track because of the weaker magnitude of ALHF over the actual track but propagated north-westward and made landfall south of the actual location. On the other hand, due to relatively strong ALHF from tuned experiments over the actual track, they crossed the observed track. They made landfall close to the actual location, resulting in a lesser track error."

**Comment 20:** 415: You have discussed the wind structure but not any storm dynamics.

**Reply:** Thank you. This comment has been addressed in revised manuscript as provided in the reply of comment no. 1.